

# Stratospheric gravity waves at southern hemisphere orographic hotspots: 2003 – 2014 AIRS/Aqua observations

Lars Hoffmann[1], Alison W. Grimsdell[2], and M. Joan Alexander[2]

[1]Jülich Supercomputing Centre, Forschungszentrum Jülich, Jülich, Germany
[2]NorthWest Research Associates, Inc., CoRA Office, Boulder, CO, USA

*Correspondence to:* L. Hoffmann (l.hoffmann@fz-juelich.de)

**Abstract.** Stratospheric gravity waves from small-scale orographic sources are currently not well-represented in general circulation models. This may be a reason why many simulations have difficulty reproducing the dynamical behaviour of the southern hemisphere polar vortex in a realistic manner. Here we discuss a 12-year record (2003 – 2014) of stratospheric gravity wave activity at southern hemisphere orographic hotspots as observed by the Atmospheric InfraRed Sounder (AIRS) aboard the National Aeronautics and Space Administration's (NASA's) Aqua satellite. We introduce a simple and effective approach, referred to as the 'two-box method', to detect gravity wave activity from infrared nadir sounder measurements and to discriminate between gravity waves from orographic and other sources. From austral mid fall to mid spring (April – October) the contributions of orographic sources to the observed gravity wave occurrence frequencies were found to be largest for the Andes (90%), followed by the Antarctic Peninsula (76%), Kerguelen Islands (73%), Tasmania (70%), New Zealand (67%), Heard Island (60%), and other hotspots (24 – 54%). Mountain wave activity was found to be closely correlated with peak terrain altitudes, and with zonal winds in the lower troposphere and mid stratosphere. We propose a simple model to predict the occurrence of mountain wave events in the AIRS observations using zonal wind thresholds at 3 hPa and 750 hPa. The model has significant predictive skill for hotspots where gravity wave activity is primarily due to orographic sources. It typically reproduces seasonal variations of the mountain wave occurrence frequencies at the Antarctic Peninsula and Kerguelen Islands from near zero to over 60% with mean absolute errors of 4 – 5 percentage points. The prediction model can be used to disentangle upper level wind effects on observed occurrence frequencies from low level source and other influences. The data and methods presented here can help to identify interesting case studies in the vast amount of AIRS data, which could then be further explored to study the specific characteristics of stratospheric gravity waves from orographic sources and to support model validation.

## 1 Introduction

Atmospheric gravity waves have substantial impact on weather and climate. They transport energy and momentum, contribute to turbulence and mixing, and influence the mean circulation and thermal structure of the middle atmosphere (Lindzen, 1981; Holton, 1982, 1983). Low frequency and long wavelength gravity waves can be explicitly resolved in mesoscale model simulations, whereas global circulation models typically require parametrization schemes to represent effects of gravity waves on subgrid-scales (Lindzen, 1981; Hines, 1997; Warner and McIntyre, 1999; Geller et al., 2013). The development of gravity





wave parametrization schemes is challenging, because gravity waves are excited by various sources, each having individual characteristics. Two prominent sources of gravity waves are orographic generation (Smith, 1979, 1985; Durran and Klemp, 1987; Nastrom and Fritts, 1992) and convection (Pfister et al., 1986; Tsuda et al., 1994; Alexander and Pfister, 1995; Vincent and Alexander, 2000). Other sources include adjustment of unbalanced flows in the jet streams and frontal systems (Fritts and

Alexander, 2003; Wu and Zhang, 2004). Another source, body forcing accompanying localized wave dissipation, is likely to occur commonly in the middle atmosphere (Vadas et al., 2003). The individual characteristics of the gravity wave sources and the alterations of the gravity wave spectrum with altitude-dependent wind and stability variations are important research topics.

In the stratosphere gravity waves from convective sources are generally most important in the summer hemisphere, where planetary wave activity is weak (Alexander and Rosenlof, 1996; Scaife et al., 2000). In the winter hemisphere orographic

and jet sources play a more important role, and small-scale orographic hotspots may provide a significant contribution to the total gravity wave drag that is currently not well-represented in global climate models (McLandress et al., 2012; Alexander and Grimsdell, 2013; Vosper, 2015). More comprehensive observations may help to develop and improve parameterizations to better incorporate the wave drag even for such small sources. In this study we analyze satellite observations of stratospheric gravity wave activity at 18 orographic hotspots located in the southern hemisphere. The study closely follows recent work

of Alexander and Grimsdell (2013), which analyzed the seasonal cycle of orographic gravity wave occurrence above remote islands in the southern oceans. Further motivation to study stratospheric gravity wave activity at mid and high latitudes during winter arises from the fact that gravity waves play an important role in the formation of polar stratospheric clouds (PSCs). Localized temperature fluctuations associated with gravity waves can yield stratospheric temperatures below the threshold values for PSC formation, even if synoptic-scale temperatures are too high (Carslaw et al., 1998; Dörnbrack et al., 1999).

Eckermann et al. (2009), Lambert et al. (2012), and Orr et al. (2015) used comprehensive satellite observations to study the impact of mountain waves at high latitudes on PSC formation.

Satellite instruments offer excellent opportunities to study gravity waves on a global scale. In this study we focus on nadir scanning observations of AIRS (Aumann et al., 2003; Chahine et al., 2006) aboard NASA's Aqua spacecraft. The main advantage of nadir sounders such as AIRS is good horizontal resolution and coverage. The disadvantage is that the nadir measurement

geometry limits the observations to gravity waves with rather long vertical wavelengths ($\lambda_z \gtrsim 15\,\mathrm{km}$ for AIRS) due to the 'observational filter' effect (Alexander, 1998; Wu et al., 2006; Alexander and Barnet, 2007). However, observations of gravity waves with long vertical and short horizontal wavelengths are of particular interest, because these waves can potentially carry large momentum flux and excite significant wave drag (Fritts and Alexander, 2003; Ern et al., 2004; Preusse et al., 2008). AIRS radiance measurements have successfully been exploited in a number of gravity wave studies. For instance, Alexander

and Teitelbaum (2007), Eckermann et al. (2007), Limpasuvan et al. (2007), Alexander and Teitelbaum (2011), Niranjan Kumar et al. (2012), and Jiang et al. (2013) demonstrated the capabilities of AIRS to observe mountain waves at orographic hotspots such as the Antarctic Peninsula, the Andes, the Greenland topography, or the Himalayas. Gong et al. (2012) and Hoffmann et al. (2013) also analyzed global long-term records of stratospheric gravity wave activity from AIRS observations. By September 2015 AIRS had completed 13 years of measurements and gathered about $13.8 \times 10^9$ infrared radiance spectra, which can be

used to explore the climatological variability of stratospheric gravity wave activity.





This study focuses on stratospheric gravity wave activity from orographic sources in the southern hemisphere, which is of particular interest in relation to the dynamical behaviour of the southern hemisphere polar vortex. The analysis is based on a 12-year record (January 2003 – December 2014) of $4.3\,\mu\text{m}$ radiance observations of AIRS/Aqua. Stratospheric gravity wave signals in terms of brightness temperature perturbations and variances are extracted by applying a number of standard

techniques developed for nadir sounders (Wu, 2004; Eckermann et al., 2006; Alexander and Barnet, 2007; Hoffmann et al., 2014). We introduce a simple and effective new method to detect orographic gravity wave signals from infrared nadir sounder measurements. To infer the orographic wave signals this method analyzes brightness temperature variance differences between two boxes located up- and downstream of an orographic hotspot. The method is used to estimate the occurrence frequencies of mountain waves at 18 orographic hotspots in the southern hemisphere based on the long-term AIRS record. Furthermore,

interactions between the mountain wave activity and tropospheric and stratospheric background winds are studied. To predict the occurrence of mountain wave events in the AIRS observations we propose a simple model based on zonal wind thresholds in the lower troposphere and in the mid stratosphere. The main purpose of the model is to provide a means of separating upper level wind effects, like the observational filter, from low level effects, like those related to the gravity wave sources. This will allow the model to be used to discuss whether waves are likely present or affecting the atmosphere even though they are only

weakly observed or invisible in the AIRS observations.

In Sect. 2 we provide a brief description of the AIRS instrument and the methods used to extract brightness temperature perturbations related to stratospheric gravity waves from the radiance measurements. In Sect. 3 we introduce the method to detect and discriminate between gravity wave signals from orographic or other sources. Seasonal mean occurrence frequencies of orographic gravity waves at various hotspots based on the 12-year AIRS record are discussed in Sect. 4. Correlations between

gravity wave activity and tropospheric and stratospheric background winds are discussed in Sect. 5. In Sect. 5 we also introduce the threshold model to predict the occurrence of mountain wave events in the AIRS observations. Section 6 focuses on inter- and intraseasonal variability of mountain wave activity at the hotspots and discusses the performance of the threshold model in explaining this variability. In Sect. 7 we provide conclusions and an outlook on how the results of this study might be used in future research.

## 2 AIRS observations of stratospheric gravity waves

AIRS (Aumann et al., 2003; Chahine et al., 2006) is one of six instruments aboard NASA's Aqua satellite. Aqua was launched in a nearly polar, low earth orbit (705 km altitude, $100°$ inclination, 100 min period) in May 2002. Nearly global coverage is achieved during 14.4 orbits per day. The Aqua orbit is sun-synchronous, with Equator crossings at 01:30 LT (descending orbit nodes) and 13:30 LT (ascending orbit nodes). AIRS measures infrared radiance spectra from the Earth's atmosphere in

the nadir and sub-limb geometry. Each across-track scan covers 1780 km ground distance and consists of 90 footprints. The scans are separated by 18 km along-track distance. The footprint size varies between $14 \times 14\,\text{km}^2$ at nadir and $21 \times 42\,\text{km}^2$ at the scan extremes. AIRS measurements cover the $3.74 – 15.4\,\mu\text{m}$ spectral range in three bands, with a resolving power of



$\lambda/\Delta\lambda = 1200$. We analyze measurements from multiple channels in the $4.3\,\mu$m spectral region, with a noise equivalent delta temperature (NEDT) of $0.13-0.15$ K at 250 K scene temperature.

We infer information on stratospheric gravity wave activity directly from the AIRS radiance measurements following the approach of Hoffmann and Alexander (2010) and Hoffmann et al. (2013, 2014). We analyze spectral mean brightness tem-
peratures in the $4.3\,\mu$m $CO_2$ fundamental band ($2322.5-2346.0$ and $2352.5-2367.0\,$cm$^{-1}$), which gets optically thick in the mid stratosphere. Temperature kernel functions for the $4.3\,\mu$m channels show a broad maximum in sensitivity of the radiances to stratospheric temperatures at $30-40$ km altitude and have a full-width at half-maximum of about 25 km (Hoffmann and Alexander, 2009; Hoffmann et al., 2014). The broad kernel functions limit the AIRS observations to gravity waves with long vertical wavelengths. We found that the 5%, 20%, and 50% response levels to wave amplitude are first exceeded at 16, 32,
and 48 km vertical wavelength, respectively (Hoffmann and Alexander, 2010; Hoffmann et al., 2014). The observed brightness temperatures are mainly composed of three contributions: (i) gravity wave signals, (ii) slowly varying background signals, and (iii) measurement noise. Background signals associated with large-scale temperature gradients or planetary waves are removed with the detrending procedure of Wu (2004), Eckermann et al. (2006), and Alexander and Barnet (2007), i. e., brightness temperature perturbations are calculated as differences from a $4^{\text{th}}$-order polynomial fit for each across-track scan. This limits
the amplitude response to 90%, 50%, and 20% at 800, 1200, and 1650 km across-track wavelength, respectively (Hoffmann and Alexander, 2010; Hoffmann et al., 2014). The short wavelength limit of the observations is at about 30 km, based on the Nyquist theorem and a sampling distance of 14 km at nadir. The noise of the spectral mean brightness temperatures is about 0.059 K at 250 K scene temperature (Hoffmann et al., 2014). The $4.3\,\mu$m brightness temperature variances shown in this paper have been corrected for noise, by subtracting noise variances scaled to scene temperature.

Climatological studies based on AIRS and other satellite observations revealed that stratospheric gravity wave activity at mid and high latitudes during the winter season is closely linked to orographic hotspots and jet sources (Gong et al., 2012; Hoffmann et al., 2013, 2014). Figure 1 shows the $2003-2014$ multi-annual seasonal mean of detrended and noise-corrected AIRS $4.3\,\mu$m brightness temperature variances in the southern hemisphere. Here we focus on the time period from mid fall to mid spring (April – October), when stratospheric gravity wave activity in the southern hemisphere is largest. Figure 1 also shows terrain
variability from a 2-minute gridded global relief data set (ETOPO2v2)[1]. The standard deviation of terrain altitudes is one of the parameters considered in gravity wave parametrization schemes for subgrid-scale orographic sources (Miller et al., 1989; Lott and Miller, 1997). The AIRS and ETOPO2v2 maps show local maxima or 'hotspots' of stratospheric gravity wave activity being clearly associated with orographic features. The strongest hotspots are found at large mountain ranges, such as the Andes, the Antarctic Peninsula, and New Zealand. Many small-scale hotspots are also evident, e. g., at some of the remote islands in
the southern oceans. The small-scale hotspots are visible due to the high horizontal resolution of the AIRS observations. Based on these maps we selected 18 hotspots of stratospheric gravity wave activity that are more closely examined in this study (Table 1). Note that some prominent hotspots at the border of East Antarctica are not considered here. In these places gravity waves are triggered by katabatic winds from mainland Antarctica (Watanabe et al., 2006), which is a rather different source mechanism from those in the other places.

---

[1]Accessible at http://www.ngdc.noaa.gov/mgg/global/etopo2.html (last access: 24 November 2015).



In addition to the orographic hotspots, the variance map in Fig. 1 shows a broad zonal band of stratospheric gravity wave activity around $50-70°$S. A pronounced maximum of gravity wave activity within this latitude band is found leeward of the Andes and the Antarctic Peninsula, extending as far as $150°$E. The origin of this broad maximum is not entirely clear as the region of enhanced activity extends well beyond the reach of the direct effect of orography. It may be caused by propagating

mountain waves (Preusse et al., 2002; Sato et al., 2012; Hindley et al., 2015), but also by non-orographic sources in winter storm tracks such as spontaneous adjustment, frontogenesis, and convection (Hendricks et al., 2014; Hindley et al., 2015; Alexander et al., 2016). Figure 2 shows $2003-2014$ April$-$October seasonal mean winds from the ERA-Interim reanalysis (Dee et al., 2011) at the AIRS observational level (3 hPa, about 40 km) and at low level (750 hPa, about 2 km). The stratospheric gravity wave activity observed by AIRS is closely linked to the winds at both levels. The activity of the orographic sources is

directly coupled to the strength of the surface winds, as strong surface winds are needed for waves to be launched. For AIRS to be able to observe the waves, strong background winds in the stratosphere are needed to foster the propagation of gravity waves with long vertical wavelengths, to which AIRS is most sensitive due to its observational filter.

## 3    Two-box method for the detection of mountain waves

In this paper we introduce a simple and effective approach, referred to as the 'two-box method', to detect gravity wave activity

at orographic hotspots from the AIRS measurements. In this method we examine the variance of detrended $4.3\,\mu$m brightness temperature perturbations in two boxes, located upstream and downstream of an orographic hotspot. We assume primarily westerly winds, so the western edge of the downstream box includes the hotspot and the box then extends to the east. The variance $\sigma_e^2$ of this box is considered to be primarily being influenced by signals from orographic gravity waves. The upstream box is located to the west of the hotspot and is not placed directly adjacent to the downstream box, but is slightly separated

to reduce the likelihood of capturing orographic wave activity in this box. Orographic waves typically propagate downstream, so the variance $\sigma_w^2$ in this box should not be affected by waves from the hotspot. The upstream box provides information on the background levels of gravity wave activity, being related to other sources. The presence of orographic wave activity is then determined from the difference in variance between these two boxes, calculated as $\sigma_{oro}^2 = \sigma_e^2 - \sigma_w^2$. The transfer of background variances from the upstream to the downstream box introduces some uncertainties in this analysis. However, large variance

differences $\sigma_{oro}^2$ most likely relate to the occurrence of orographic waves. We cope with the uncertainties of the method by introducing a variance threshold $\sigma_0^2$ and by considering only those events exceeding the threshold, $\sigma_{oro}^2 \geq \sigma_0^2$, as being related to the orographic source. Note that we applied the method to noise-corrected brightness temperature variances $\sigma_e^2$ and $\sigma_w^2$, but due to the difference approach it also bears the potential to provide effective noise correction itself.

Figure 3 shows examples of orographic wave events at selected hotspots detected with the two-box method. The events

shown here are among those with the largest $\sigma_{oro}^2$ values that we found in the 12-year record of AIRS data and in all cases the wave patterns clearly indicate orographic wave activity at the hotspots. As can be seen from the maps, we have chosen the box positions and sizes individually for each hotspot. Common box sizes for all hotspots may be desirable, in principle, regarding the wavelength sensitivities of the method. However, we found that individual optimization of the box sizes to the typical





size of the wave patterns at the hotspots improves the detection rates. Large boxes were used for strong hotspots producing extensive wave patterns such as the Andes and New Zealand (with box sizes of $10° \times 8°$ in longitude $\times$ latitude) and the Antarctic Peninsula ($15° \times 5°$). Mid-size boxes were used for Kerguelen and Tasmania ($6° \times 6°$) as well as Crozet and South Georgia ($5° \times 5°$). Small boxes were used for Balleny and Peter I Island ($5° \times 3°$), located at high latitudes, and the remaining

hotspots from Table 1 ($3° \times 3°$), located at mid latitudes. For most of the hotspots the mean latitude of the boxes was chosen to match the latitude of the hotspot. However, for Heard we applied a latitudinal shift, to stay away from orographic waves created at Kerguelen (cf. Fig. 3). The longitudinal separation between the western and eastern boxes was $1°$ for all hotspots.

In order to validate the two-box method we performed two tests that compare the results of this automatic detection of orographic wave events with statistics from visual inspection of AIRS brightness temperature maps. For the first test we used

the automatic detection method to select the three events for each year and each hotspot which had the largest $\sigma^2_{oro}$ values. This gave us $3 \times 12 \times 18 = 648$ individual events, for which we inspected the AIRS images to verify that orographic wave activity was visible at the hotspot. The performance of the two-box method varied between the hotspots. The largest success rates were found for the Andes (100%) and the Antarctic Peninsula (100%), followed by Kerguelen (93%), New Zealand (91%), Balleny (88%), Heard (78%), South Georgia (77%), and Tasmania (74%). For the remaining hotspots the success rates were below

54% and became as low as 6% for Macquarie. This test indicates that the two-box method performs best for strong hotspots with frequent wave activity and large values of $\sigma^2_{oro}$. The success rates clearly correlate with the peak altitude of the hotspots (see Table 1). Results for weak hotspots with low success rates should be considered more carefully, because those are more likely to be influenced by gravity waves from non-orographic sources.

As a second test we compared the detection results from the two-box method with the event statistic of Alexander and

Grimsdell (2013). The study of Alexander and Grimsdell (2013) analyzed gravity wave activity at Auckland, Heard, Kerguelen, Prince Edward, South Georgia, and Tasmania during the years 2003 and 2004. Orographic wave events were identified by visual inspection of AIRS $15 \, \mu m$ brightness temperature perturbation maps. The vertical coverage of the AIRS channel analyzed in that study ($667.8 \, \text{cm}^{-1}$) is at slightly higher altitudes (around $35-45$ km) than in this work. Noise levels of the $15 \, \mu m$ data are about a factor of 7 larger than the $4.3 \, \mu m$ data used here. This introduces some uncertainty when we compare the detection

results. Considering the data of Alexander and Grimsdell (2013) as 'observations' and the results from the two-box method as 'predictions', we calculated a set of skill scores (Schaefer, 1990; Wilks, 2011) to assess the performance of the two-box method for two choices of the variance threshold, $\sigma^2_0 = 0.1 \, \text{K}^2$ and $\sigma^2_0 = 1 \, \text{K}^2$, to detect orographic wave activity. These values define a reasonable range of thresholds. The total number of events decreases significantly for thresholds much larger than $1 \, \text{K}^2$. Choosing thresholds much lower than $0.1 \, \text{K}^2$, this would include many events with rather low wave amplitudes that may

not be too important overall. Using a variance threshold of $\sigma^2_0 = 1 \, \text{K}^2$, we found a bias (ratio of predictions/observations) of 12%, a probability of detection (POD) of 11%, a false alarm rate (FAR) of 5%, and a Gilbert skill score (GSS) of 7%. With such a large variance threshold the two-box method missed many of the weaker events identified by Alexander and Grimsdell (2013). This leads to a low POD, but also to a good FAR. The fact that the GSS is larger than zero indicates that the method still does have skill compared to a random forecast. Choosing a threshold of $\sigma^2_0 = 0.1 \, \text{K}^2$ improves the skill scores of the

method substantially. The bias is then 69%, the POD is 57%, the FAR is 18%, and the GSS is 33%. Future work may focus





on fine-tuning of the variance threshold, including possible optimization for the individual hotspots. However, for this study we decided to focus on events characterized by a globally constant variance threshold to allow us to compare the results of the different hotspots to each other. We selected $\sigma_0^2 = 0.1\,\mathrm{K}^2$ since with this value more events are included and the method has better skill than when using $\sigma_0^2 = 1\,\mathrm{K}^2$.

## 4 Seasonal mean occurrence frequencies of mountain waves

In this section we discuss the seasonal mean occurrence frequencies of stratospheric gravity waves at the orographic hotspots. As a first step we calculated histograms of the variances in the eastern and western boxes at each hotspot (Fig. 4). In these histograms increased numbers of events in the eastern boxes point to more frequent orographic wave activity. To quantify the increase, we calculated the ratio $n_e/n_w$ of the numbers of events below and above the identity line, respectively. A large ratio $n_e/n_w$ indicates that the occurrence of orographic waves is more likely. Figure 4 and Table 1 show that $n_e/n_w$ is largest for the Andes (20.8), followed by the Antarctic Peninsula (6.9), Kerguelen (6.0), Heard (3.4), Prince Edward (3.4), Tasmania (2.8), South Georgia (2.6), New Zealand (2.4), Auckland (2.3), Balleny (2.1), and Tristan (2.1). For most of the remaining hotspots $n_e/n_w$ ranges from 1.3 to 1.9, indicating that orographic wave activity is less frequent. For South Orkney the ratio is 0.9, i. e., gravity wave activity in the upstream (western) box exceeds that in the downstream (eastern) box. This is due to the western box often being influenced by orographic waves from the Antarctic Peninsula. In principle, the ratio $n_e/n_w$ provides a simple way to select hotspots that are well suited to study orographic wave activity. However, this selection can be further optimized by considering uncertainties in measurement coverage and uncertainties in the confidence level at which orographic waves are detected, as will be discussed below.

The total number of events in the histograms in Fig. 4 depends on the number of satellite overpasses at each hotspot. Usually there are two satellite overpasses per day at each location, but this varies with latitude. At the equator there are regular data gaps between the AIRS swaths from neighbouring overpasses, these gaps become narrower with increasing latitude. The AIRS swaths start to overlap at $\pm 45°$ latitude. At high latitudes there is significant overlap of the swaths so that there may be four, or even more, overpasses per day, which can be analyzed. The area observed during an overpass varies with each orbit and does not always cover the entire area of a box. In our analysis we considered only those overpasses that covered at least 50% of the area of the boxes, to ensure the variance calculations were robust. To cope with the variability in measurement coverage, we focus on occurrence frequencies, i. e., fractions of overpasses showing gravity wave activity with respect to the total number of overpasses, rather than event counts in the rest of the paper. Table 1 provides the $2003-2014$ April$-$October seasonal mean occurrence frequencies $f_{gw}$ of all observed gravity waves at each hotspot. These were calculated using only the information in the eastern box, and applying a variance threshold of $\sigma_e^2 \geq 0.1\,\mathrm{K}^2$. The occurrence frequencies $f_{gw}$ vary greatly between the hotspots, from 4.8% for Tristan to 59% for the Andes. Table 1 also presents the occurrence frequency $f_{oro}$ of orographic waves determined with the two-box method with $\sigma_0^2 = 0.1\,\mathrm{K}^2$ for each hotspot. The occurrence frequencies $f_{oro}$ vary from 1.5% for Tristan to 53% for the Andes. Furthermore, Table 1 presents the ratio $f_{oro}/f_{gw}$ as a measure of the contribution of orographic wave activity to total gravity wave activity as observed by AIRS at each hotspot. The ratio $f_{oro}/f_{gw}$ varies from 24% for Gough





to 90% for the Andes. It is important to note that the absolute values of $f_{gw}$ and $f_{oro}$ largely depend on the choice of $\sigma_0^2$. For instance, raising $\sigma_0^2$ from 0.1 to $1\,K^2$, $f_{gw}$ and $f_{oro}$ typically decrease by a factor of $5-15$. However, while the frequencies changed, the ratio $f_{oro}/f_{gw}$ remained nearly constant and we found that the rankings between different hotspots in terms of any of these measures – $f_{oro}$, $f_{gw}$, and the ratio $f_{oro}/f_{gw}$ – are largely independent of the choice of $\sigma_0^2$. The ratio $f_{oro}/f_{gw}$ provides

a good way to select hotspots that are best suited to study orographic wave activity.

Note that the occurrence frequency $f_{oro}$ of orographic waves is expected to grow with the terrain peak altitude, because taller mountains cause larger vertical displacements in the flow and so will generate larger amplitude waves. A statistical association between terrain peak altitudes and gravity wave occurrence frequencies was also found from Table 1. The peak altitudes listed here are local maxima of the ETOPO2v2 data in the eastern boxes considered for the two-box method. The

maxima are representative for 2-minute horizontal grid resolution and can be lower than actual mountain peak heights (cf. Table 1 of Alexander and Grimsdell, 2013). We calculated the Spearman rank-order correlation coefficient $\rho_s$ between the terrain peak altitudes and the seasonal mean occurrence frequencies $f_{gw}$ and $f_{oro}$ of the hotspots. We found a medium degree of correlation ($\rho_s = 0.39$) using $f_{gw}$ and a high degree of correlation ($\rho_s = 0.70$) using $f_{oro}$. This indicates that the two-box method effectively identifies orographic wave events, for which occurrence frequencies are closely linked to terrain altitude.

Another important factor controlling the occurrence frequencies $f_{gw}$ and $f_{oro}$ are the background winds in the troposphere and stratosphere, which will be discussed in the next section.

## 5  Correlations of mountain wave activity and background winds

### 5.1  Mountain wave characteristics from linear wave theory

In this section we analyze correlations between the orographic wave occurrence frequencies and the background winds at

the hotspots. However, we first repeat some of the typical characteristics of mountain waves and their relationships to the background winds as inferred from linear wave theory (e. g., Smith, 1979; Fritts and Alexander, 2003; Holton and Hakim, 2012). Starting from the dispersion relation for gravity waves with midrange intrinsic frequencies,

$$\hat{\omega} = \frac{N\,k}{m}, \tag{1}$$

with intrinsic frequency $\hat{\omega} = k\,u$ for mountain waves, buoyancy frequency $N$, horizontal wavenumber $k$, and vertical wavenum-

ber $m$, it can be shown that the vertical wavelength $\lambda_z = 2\pi/m$ is linearly proportional to the background wind $u$,

$$\lambda_z = \frac{2\pi}{N}u. \tag{2}$$

For instance, in the troposphere ($N \approx 0.01\,s^{-1}$) a background wind $u = 10\,m\,s^{-1}$ triggers gravity waves with $\approx 6\,km$ vertical wavelength. In the stratosphere the restoring force is stronger, increasing the buoyancy frequency ($N \approx 0.02\,s^{-1}$) and potentially reducing the vertical wavelength. However, in the mid- and high-latitude austral winter the stratospheric background winds are

much stronger than the low-level winds (up to a factor of $5-10$), which typically shifts the vertical wavelengths into a range observable by AIRS, despite the opposing effect of the increased buoyancy.





As a representative example, we performed 2-D gravity wave raytracing calculations for ERA-Interim monthly mean temperature and zonal wind profiles at the Kerguelen Islands (Fig. 5). The raytracing model and some of its applications are described in more detail by Alexander (1998) and Yue et al. (2013, 2014). The model uses a more general form of the gravity wave dispersion relation and is not limited to mid-frequency waves. In our simulations the launch level of the mountain waves was set to 2 km. The horizontal wavelength was set to 100 km, which is a typical horizontal wavelength from AIRS observations. The raytracing calculations show that gravity waves are launched with a typical vertical wavelength of $\approx 9\,\mathrm{km}$ in the troposphere, due to mean zonal winds of $\approx 18\,\mathrm{m\,s^{-1}}$ and buoyancy frequencies of $\approx 0.012\,\mathrm{s^{-1}}$ at the launch level. Mean westerly winds foster the propagation of gravity waves through the stratosphere from April to September. A critical layer related to easterly winds at 40 km was found in October. At the AIRS observational levels (about 40 km) vertical wavelengths become as large as $20-30\,\mathrm{km}$ from June to August, which is well within the range of wavelengths observable by AIRS. The mountain waves have large group velocities in the stratosphere (about $10-30\,\mathrm{m/s}$), yielding short horizontal propagation distances ($5-10\,\mathrm{km}$) and propagation times ($90-150\,\mathrm{min}$) to reach the mid stratosphere.

## 5.2 Time series and correlation analyses of gravity wave activity and background winds

In this section we discuss time series of the orographic gravity wave variances $\sigma_{oro}^2$ based on individual AIRS overpasses and ERA-Interim background winds $(u, v)$ at different height levels above the hotspots. As an example, Fig. 6 shows time series of $\sigma_{oro}^2$ and $u$ at the Kerguelen Islands. The years 2005, 2006, and 2007 shown here are characterized by a low, high, and medium level of gravity wave activity, respectively. The values of $u$ are area averages for the eastern box and refer to the AIRS observational level (3 hPa, about 40 km) and low level (750 hPa, about 2 km). We linearly interpolated in time from the 6-hourly ERA-Interim data to the measurement times of the AIRS/Aqua overpasses. Figure 6 also provides the Spearman rank-order correlation coefficient $\rho_s$ between $\sigma_{oro}^2$ and $u$ at the two height levels. Although vertical wavelengths scale linearly with the background wind to first order according to Eq. (2), the sensitivity of AIRS to different vertical wavelengths is non-linear. Therefore rank-order correlation coefficients instead of Pearson's linear correlation coefficients are analyzed here (Wilks, 2011). For the example of Kerguelen Islands and the years from 2005 to 2007 in Fig. 6 we found a high degree of correlation with the zonal wind at the observational level, with $\rho_s(40\,\mathrm{km})$ ranging from 0.78 to 0.84. These large correlation coefficients indicate that the observations are strongly influenced by the observational filter that is controlled by the background wind at the height level of the observations. We found a weak degree of correlation at low level, with $\rho_s(2\,\mathrm{km})$ ranging from 0.21 to 0.27. This indicates that although the influence of orographic sources on the observations is weaker than that of the upper level winds, information on the orographic sources is still present in the measurements.

We performed correlation analyses of the AIRS and ERA-Interim time series for the years $2003-2014$ for the first nine hotspots listed in Table 1. At these hotspots the gravity wave activity is primarily due to the orographic sources, $f_{oro}/f_{gw} \gtrsim 50\%$. Figure 7 shows vertical profiles of $\rho_s$ with respect to the zonal and meridional winds at different altitudes for the Antarctic Peninsula and Kerguelen. The results for the other hotspots are similar. Here we selected the Antarctic Peninsula and Kerguelen as representative examples of a mountain ridge and a peak, respectively. For a mountain ridge it may be expected that orientation of the background winds with respect to the ridge may also play a role, as waves are best formed parallel to the ridge and





perpendicular to the wind (Hines, 1988). Figure 7 shows mean and standard deviation profiles of $\rho_s$ based on individual years. Standard deviations are mostly in the range of $0.1 - 0.2$, indicating that the interannual variations of $\rho_s$ are small. Regarding correlations with the zonal winds (black curves in Fig. 7), we found a high degree of correlation for a broad maximum in the mid and upper stratosphere ($\rho_s$ up to $0.6 - 0.8$ at $30 - 50$ km altitude), reflecting the influence of the AIRS observational filter. A

second, weaker maximum of correlations was found in the lower troposphere ($\rho_s$ up to $0.2 - 0.4$ at 2 km altitude), reflecting the influence of the orographic sources. Correlations typically are at a minimum in the upper troposphere and lower stratosphere (near 10 km altitude). Regarding correlations with the meridional winds (gray curves in Fig. 7), strong correlations are generally not expected at peaks such as Kerguelen, because tropospheric and stratospheric winds are predominantly westerly (Fig. 2). Some degree of correlation could be expected for the Antarctic Penisula, with the mountain ridge being aligned from southwest

to northeast. However, we found that correlations with the low level meridional winds are low in both cases, with $\rho_s$ being below $\pm 0.1$. At stratospheric levels the correlations with the meridional wind became larger, but $\rho_s$ typically still did not exceed levels of $\pm 0.2$.

Note that the zonal and meridional winds in the troposphere or stratosphere are dynamically coupled and therefore strongly correlated. The correlations between the gravity wave activity and the zonal and meridional winds found here are therefore

directly linked to the correlations of the winds themselves. To illustrate this, we calculated the correlations of the zonal winds at 2 km and 40 km altitude with the meridional and zonal winds at other height levels. From Fig. 7 it can be seen that the zonal winds have rather large correlation lengths in the vertical domain. The vertical correlations of the 40 km zonal wind steadily decrease toward zero at the 10 km height level. The vertical correlations of the 2 km zonal wind fade away at 25 km altitude for Kerguelen and 40 km altitude for the Antarctic Peninsula. In Fig. 7 it can also be seen that anticorrelations (Antarctic

Peninsula) or correlations (Kerguelen) of the gravity wave activity with respect to the meridional wind are directly related to anticorrelations or correlations between the meridional and zonal wind components. Based on this correlation analysis we concluded that the zonal wind provides a good proxy for the total background wind activity on its own. It is largely sufficient to analyze the zonal winds at the two height levels (2 km and 40 km) selected here, which provide independent information.

We performed another correlation analysis to demonstrate that the background wind data can be used to effectively disen-

tangle upper level wind effects on the AIRS gravity wave observations from low level source and other influences. Due to the observational filter the AIRS observations are limited to gravity waves with long vertical wavelengths, which in turn require strong background winds at the observational level (Sect. 5.1). In order to reduce the influence of the observational filter, we performed the correlation analysis only for those events in the AIRS time series, for which the 40 km zonal winds exceed selected thresholds. Here we selected zonal wind thresholds of 44 m/s for the Antarctic Peninsula and 72 m/s for the Kerguelen

Islands. These thresholds are also applied in the prediction model for mountain wave events, which will be introduced in more detail in Sect. 5.3. For the filtered AIRS time series including only cases with strong upper level winds (green curves in Fig. 7) we found that the correlations with the low level winds increased whereas correlations with upper level winds decreased. The correlation coefficient $\rho_s(2\,\mathrm{km})$ increased from 0.39 to 0.54 for the Antarctic Peninsula and from 0.23 to 0.42 for the Kerguelen Islands. In contrast, $\rho_s(40\,\mathrm{km})$ decreased from 0.59 to 0.35 for the Antarctic Peninsula and from 0.81 to 0.27 for the Kerguelen

Islands. This shows that the focus on events with strong upper level winds provides an efficient method to compile AIRS time





series that more directly provide information on the gravity wave sources at lower levels. This approach is pursued further in a prediction model for mountain wave events based on wind thresholds that will be introduced next.

### 5.3 Prediction model for mountain wave events based on wind thresholds

In this section we introduce a simple model that can be used to predict the occurrence of mountain wave events at orographic hotspots in the AIRS observations based on the zonal winds in the lower troposphere and mid stratosphere. Mountain waves are launched when there are strong winds near the surface. Strong background winds at higher altitudes are required to foster the propagation of gravity waves with long vertical wavelengths into the stratosphere (Sect. 5.1). We present a binary model that can be used to reliably predict the occurrence of a mountain wave event in the AIRS observations if the zonal winds $u(z)$ at $z = 2\,\mathrm{km}$ and $z = 40\,\mathrm{km}$ both exceed given thresholds, $u_0(z)$. A skill score analysis was performed to establish these zonal wind thresholds. Binary observations of orographic waves are based on the variance threshold criterion, $\sigma_{oro}^2 \geq \sigma_0^2$ with $\sigma_0^2 = 0.1\,\mathrm{K}^2$, as introduced in Sect. 3. We calculated the GSS of this prediction model for zonal wind thresholds between $-20\,\mathrm{m/s}$ and $120\,\mathrm{m/s}$ in steps of $1\,\mathrm{m/s}$. The results for the Antarctic Peninsula and the Kerguelen Islands are presented in Fig. 8. In most cases the GSS distributions showed a clear maximum (e. g., for the Antarctic Peninsula in Fig. 8). We found that the GSS distributions are tightly constrained by the winds at the observational level (40 km) whereas the low level winds (2 km) seem to play a smaller role. This is similar to results of Alexander and Grimsdell (2013), who found that the surface winds at southern hemisphere orographic hotspots are generally strong enough to generate gravity waves so the stratospheric winds were a better predictor of wave observations in AIRS. Exceptions occurred when surface winds blew westward, a situation that prevents any waves generated from penetrating to upper levels. For a few hotspots we found that the low level winds did not help to identify a clear maximum (e. g., for Kerguelen Islands in Fig. 8). To cope with this issue and to estimate uncertainty, we determined the wind thresholds from the upper 5% percentile of the GSS distributions. As an additional constraint, we considered only data points with a bias in the range of $90 - 110\%$, so that the model is not significantly under- or overforecasting the total number of events.

The results of the skill score analysis are summarized in Table 2. Again, we focus on those nine hotspots where gravity wave activity is primarily related to orographic sources ($f_{oro}/f_{gw} \gtrsim 50\%$). GSS values in the range of $26 - 42\%$ indicate that the prediction model has good skill. The model has no significant biases ($98 - 103\%$), good PODs ($59 - 80\%$), and mostly low FARs ($18 - 43\%$). We found that the wind thresholds at the GSS maxima vary substantially between the hotspots, i. e., between $44\,\mathrm{m/s}$ (Andes) and $80\,\mathrm{m/s}$ (Prince Edward) at the $40\,\mathrm{km}$ level and $3\,\mathrm{m/s}$ (Antarctic Peninsula) and $18\,\mathrm{m/s}$ (Prince Edward) at the $2\,\mathrm{km}$ level. Among the most important factors influencing the thresholds are the different peak terrain altitudes and the background winds at the hotspots (cf. Fig. 2). Another factor influencing the thresholds in the case of mountain ridges could be the orientation of the winds with respect to the ridge. However, the correlation analysis presented in Sect. 5.2 suggests that this is a second-order effect. Nevertheless, the wind ranges found here are generally consistent with theory. A range of low level winds of about $5 - 15\,\mathrm{m/s}$ is best suited for wave generation, because weak or westward winds would give weak or no waves, whereas very strong eastward winds are associated with instability. Stratospheric background winds greater than $40\,\mathrm{m/s}$ clearly foster the propagation of waves with vertical wavelengths visible to AIRS (Sect. 5.1).



We also tested the sensitivity of the skill score analysis regarding the variance threshold $\sigma_0^2$ used to detect orographic wave activity. Increasing $\sigma_0^2$ to $1\,K^2$, we found that the GSS decreased by $5-10$ percentage points. This indicates that the prediction model has lower skill for predicting the occurrence of the strongest wave events. However, note that such strong events appear very infrequently, so the statistical sample size is substantially reduced, and individual observational effects are

more influential. Decreasing $\sigma_0^2$ to $0.01\,K^2$, we found that GSS increased by $5-10$ percentage points. However, with this low threshold a large number of rather weak events is included, which may not contribute significantly to gravity wave drag or that are not even related to orographic sources at all. Note that the analysis for the remaining nine hotspots of Table 1 (Auckland to Gough) for our default threshold of $0.1\,K^2$ yields lower skills (GSS range of $10-24\%$), which was expected as orography is not the leading source mechanism in these places. The model is only applicable for orographic hotspots.

Figure 9 shows histograms of the $2003-2014$ April–October ERA-Interim zonal winds at the $2\,km$ and $40\,km$ height levels for the Antarctic Peninsula and Kerguelen Islands. In the analysis of the wind distributions we considered two cases. In the first case we used wind data from all satellite overpasses over the hotspots, whereas in the second case we considered only data from overpasses with AIRS showing orographic wave events. The overall wind distributions (first case, light colors in Fig. 9) typically cover broad ranges of easterlies and westerlies, with 90% of the events being located in zonal wind ranges of about

$-10$ to $30\,m/s$ at the $2\,km$ level and about $-20$ to $110\,m/s$ at the $40\,km$ level. At the observational level the hotspots at mid latitudes (e.g., Kerguelen) have rather broad and flat distributions. The hotspots at high latitudes (e.g., Antarctic Peninsula) show a pronounced zonal wind maximum at 30 to $80\,m/s$ due to the polar jet. The orographic wave events (second case, dark colors in Fig. 9) are associated with strong westerly winds, most notably at the observational level (with zonal wind ranges shifted to 40 to $120\,m/s$), but also at low level (with zonal wind ranges shifted to 0 to $30\,m/s$). Wind reversals from westerlies

to easterlies prohibit the propagation of gravity waves into the stratosphere. Consequently, no wave events associated with easterlies at the $40\,km$ level are found in the wind distributions. Regarding the $2\,km$ level, we found that about 2.8% (Antarctic Peninsula) and about 0.3% (Kerguelen) of the events are associated with easterlies. These few outliers are likely due to false detections of orographic wave events with the two-box method as well as uncertainties of the ERA-Interim winds. Similar to the skill score analysis presented earlier, the analysis of wind distributions suggests that the wind distributions associated with

mountain wave events are clearly affected by the overall wind distributions. Other factors such as the orientation of a mountain ridge with respect to the mean wind direction may contribute. This causes the wind thresholds of the prediction model to vary between the different hotspots. They need to be tuned for each location. Note that we also indicated the wind thresholds of the prediction model in Fig. 9. This shows that large fractions ($60-80\%$) of the observed mountain wave events are in fact covered by the model.

## 6 Inter- and intraseasonal variations of mountain wave activity

In this section we discuss yearly and monthly variations of the orographic wave activity at the hotspots. We focus on results for the Antarctic Peninsula and Kerguelen Islands, being representative examples of a mountain ridge and a peak, respectively. Figure 10 shows $2003-2014$ monthly mean occurrence frequencies of the orographic waves from AIRS observations and the



prediction model. In addition, Fig. 10 shows monthly occurrence frequencies of the ERA-Interim zonal winds exceeding the thresholds at the 2 km and 40 km levels at the hotspots. The occurrence frequencies of the zonal winds were calculated using the wind thresholds defined in Sect. 5.3 and Table 2. A clear seasonal variation is found in the monthly occurrence frequencies, with minima of 1 – 12% in April and October and maxima as large as 62% in July. For the Antarctic Peninsula we found a

rather long season, with occurrence frequencies exceeding the 50% level from May to September. At Kerguelen the 50% level is exceeded only in June and July. The intraseasonal variations of the orographic waves clearly follow the occurrence frequencies of the zonal winds at the observational level. Those cover ranges of 14 – 96% at the Antarctic Peninsula and 0 – 88% at the Kerguelen Islands. For both the Antarctic Peninsula and Kerguelen the occurrence frequencies of the low level winds are in the range of 60 – 80% from April to October on average, which indicates a high chances for orographic waves being excited at all

times. The prediction model reproduces the monthly variations of the observed occurrence frequencies with a mean absolute error of 5 percentage points. A larger error of 15 percentage points was found only for Kerguelen Island in June, and seems to be related to an overestimation of the influence of the observational level wind on the wave activity.

Figure 10 also presents the seasonal mean occurrence frequencies at the Antarctic Peninsula and Kerguelen Islands for individual years from 2003 to 2014. The time series reveal substantial interannual variations of the occurrence frequencies,

covering ranges of 33 – 52% at the Antarctic Peninsula and 10 – 38% at Kerguelen Islands. The annual variations of the gravity wave occurrence frequencies are again found to be closely correlated with the occurrence frequencies of the zonal winds. The example for the Antarctic Peninsula indicates that even though winds at the observational level are often most influential, the low level winds are still important. This becomes most evident during the years 2005 to 2010. Given that the wind at the observational level remains high during this time, the occurrence of gravity waves then clearly follows the low level winds. This

shows that both levels are required to predict AIRS observations of orographic waves. The prediction model reproduces the interannual variations of the seasonal occurrence frequencies with a mean absolute error of 4 percentage points. The absolute errors became as large as 12 percentage points (Antarctic Peninsula) and 9 percentage points (Kerguelen) in individual years. The large interannual variability indicates that to get statistically meaningful results the occurrence frequencies should be calculated based on long-term records such as those provided by AIRS.

**7 Conclusions**

In this paper we introduced the simple and effective two-box method that can be used to detect orographic gravity waves in infrared nadir sounder imagery. The method was applied to 12 years of AIRS/Aqua observations to analyze mountain wave activity during April to October at 18 orographic hotspots in the southern hemisphere. The seasonal mean mountain wave activity was most frequent over the Andes (with an occurrence frequency of 53%), followed by the Antarctic Peninsula (43%),

Kerguelen (25%), South Georgia (24%), Heard (23%), Balleny (17%), and less than 13% in other places. At many hotspots mountain waves contribute significantly to the total gravity wave activity as observed by AIRS. Contributions are as large as 90% at the Andes, followed by the Antarctic Peninsula (76%), Kerguelen (73%) Tasmania (70%), New Zealand (67%), Heard (60%), and other hotspots (24 – 54%). Mountain wave occurrence frequencies are closely correlated with terrain peak altitudes



($\rho_s = 0.70$). Orographic gravity wave variances are also strongly correlated with the zonal background wind at 40 km altitude (with $\rho_s$ varying between 0.6 and 0.8), which is attributed to the AIRS observational filter. Weaker correlations are found with respect to low level winds at 2 km altitude (with $\rho_s$ varying between 0.2 and 0.4), but this may be mostly due to the fact that the low level winds at the hotspots were rarely below the threshold required for launching waves. We developed a simple model

that predicts the occurrence frequencies of mountain waves in AIRS observations based on zonal wind thresholds at 2 km and 40 km altitude. This prediction model has significant skill (GSS of $10-42\%$). It reproduces yearly and monthly variations of the mountain wave occurrence frequencies at the Antarctic Peninsula and Kerguelen which vary from near zero to over 60% with mean absolute errors of $4-5$ percentage points.

Our results on the seasonal cycle of gravity wave activity at southern hemisphere hotspots and correlations with the back-

ground winds agree well with those by Alexander and Grimsdell (2013). Use of the orographic wave detection algorithm developed here, and the thresholds found for upper and lower level wind, could permit extension of their wave flux estimates to include geographic and interannual variability more comprehensively. This would allow us to better characterize the collective effect of these waves on the circulation of the southern hemisphere stratosphere. Although terrain peak altitudes and zonal background winds are most closely correlated with mountain wave occurrence frequencies, there are many other factors that

influence the excitation, propagation, and observability of these waves. These include: (i) source variations such as terrain roughness, slope, and orientation with respect to the surface winds, (ii) wind variations between different height levels and between the zonal and meridional components, and (iii) observational effects related to the AIRS measurement geometry, e. g., the orientation of the wave fronts with respect to the line of sight. Future work should aim for improved understanding of these effects. The two-box method and the prediction model based on wind thresholds introduced here can be used to identify

interesting case studies in the vast amount of AIRS data, improving the usefulness of the data for future research on mountain waves and their impact on atmospheric dynamics.

*Acknowledgements.* AIRS data products are distributed by the NASA Goddard Earth Sciences Data Information and Services Center. ERA-Interim data are provided by the European Centre for Medium-Range Weather Forecasts. The ETOPO2v2 data set was obtained from the U.S. Department of Commerce, National Oceanic and Atmospheric Administration, National Geophysical Data Center. Support for AG and

MJA was provided by the Science of Terra and Aqua program, NASA contract #NNH11CD34C, and additional support for MJA by the NASA Shared Services Center, grant #NNX14A076G. We thank Catrin Meyer for comments on an earlier draft of this manuscript.





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





**Table 1.** Southern hemisphere orographic hotspots of stratospheric gravity wave activity. In this table $n_e/n_w$ refers to the ratio of events with gravity waves variances in the eastern box being larger than in the western box, $f_{gw}$ to the gravity wave occurrence frequency, and $f_{oro}$ to the orographic wave occurrence frequency as observed by AIRS. The table is ordered by the ratio $f_{oro}/f_{gw}$.

| Hotspot | Latitude | Longitude | Altitude [m] | $n_e/n_w$ | $f_{gw}$ [%] | $f_{oro}$ [%] | $f_{oro}/f_{gw}$ [%] |
|---|---|---|---|---|---|---|---|
| Andes | 50.0°S | 77.0°W | 4405 | 20.8 | 59.1 | 52.8 | 89.4 |
| Antarctic Peninsula | 65.0°S | 70.0°W | 2236 | 6.9 | 56.0 | 42.7 | 76.3 |
| Kerguelen | 49.3°S | 68.6°E | 1792 | 6.0 | 34.4 | 25.4 | 73.9 |
| Tasmania | 41.9°S | 144.5°E | 1490 | 2.8 | 11.1 | 7.8 | 70.2 |
| New Zealand | 44.0°S | 166.5°E | 2983 | 2.4 | 13.5 | 9.1 | 67.3 |
| Heard | 54.1°S | 73.2°E | 2192 | 3.4 | 36.3 | 21.9 | 60.3 |
| South Georgia | 54.2°S | 38.1°W | 1831 | 2.6 | 44.1 | 23.8 | 54.0 |
| Prince Edward | 46.9°S | 37.6°E | 964 | 3.4 | 23.1 | 12.4 | 53.6 |
| Balleny | 67.0°S | 162.1°E | 1352 | 2.1 | 34.3 | 17.1 | 49.8 |
| Auckland | 50.7°S | 166.1°E | 403 | 2.3 | 14.6 | 6.4 | 43.7 |
| Peter I | 68.8°S | 90.8°W | 1328 | 1.5 | 21.1 | 7.3 | 34.6 |
| Crozet | 46.4°S | 50.1°E | 599 | 1.9 | 17.0 | 5.8 | 33.9 |
| South Sandwich | 58.4°S | 26.5°W | 903 | 1.3 | 35.8 | 12.0 | 33.6 |
| Tristan | 37.1°S | 12.4°W | 1344 | 2.1 | 4.8 | 1.5 | 32.1 |
| Macquarie | 54.6°S | 158.8°E | 206 | 1.4 | 16.3 | 5.2 | 32.0 |
| Bouvet | 54.4°S | 3.3°E | 298 | 1.4 | 25.3 | 7.4 | 29.2 |
| South Orkney | 60.6°S | 45.5°W | 755 | 0.9 | 41.2 | 11.0 | 26.7 |
| Gough | 40.3°S | 10.1°W | 758 | 1.7 | 7.9 | 1.9 | 23.7 |





**Table 2.** Zonal wind thresholds and skill scores of the mountain wave prediction model. The table provides the probability of detection (POD), the false alarm rate (FAR), and the Gilbert skill score (GSS).

| Hotspot | $u_0(40\,\mathrm{km})$ [m/s] | $u_0(2\,\mathrm{km})$ [m/s] | bias [%] | POD [%] | FAR [%] | GSS [%] |
|---|---|---|---|---|---|---|
| Andes | $50 \pm 3$ | $6 \pm 2$ | $98 \pm 5$ | $80 \pm 3$ | $18 \pm 2$ | $39.5 \pm 0.5$ |
| Antarctic Peninsula | $44 \pm 3$ | $3 \pm 2$ | $99 \pm 5$ | $73 \pm 3$ | $26 \pm 1$ | $31.3 \pm 0.4$ |
| Kerguelen | $72 \pm 3$ | $13 \pm 3$ | $101 \pm 5$ | $72 \pm 2$ | $29 \pm 1$ | $41.6 \pm 0.7$ |
| Tasmania | $69 \pm 2$ | $7 \pm 2$ | $102 \pm 4$ | $63 \pm 2$ | $38 \pm 1$ | $40.0 \pm 0.4$ |
| New Zealand | $64 \pm 2$ | $6 \pm 2$ | $100 \pm 5$ | $62 \pm 2$ | $38 \pm 1$ | $38.8 \pm 0.6$ |
| Heard | $72 \pm 3$ | $12 \pm 2$ | $101 \pm 6$ | $67 \pm 3$ | $34 \pm 1$ | $35.3 \pm 0.4$ |
| South Georgia | $73 \pm 2$ | $6 \pm 3$ | $102 \pm 5$ | $62 \pm 2$ | $39 \pm 1$ | $26.0 \pm 0.3$ |
| Prince Edward | $80 \pm 2$ | $18 \pm 2$ | $103 \pm 5$ | $59 \pm 2$ | $43 \pm 1$ | $32.3 \pm 0.4$ |
| Balleny | $50 \pm 3$ | $12 \pm 2$ | $98 \pm 6$ | $60 \pm 2$ | $39 \pm 1$ | $30.5 \pm 0.4$ |



**Figure 1.** Top: Multi-annual seasonal mean (April – October in 2003 – 2014) of detrended and noise-corrected AIRS 4.3 µm brightness temperature variances due to stratospheric gravity wave activity. Bottom: Terrain altitude standard deviations from 2-minute gridded global relief data (ETOPO2v2) at 0.25° × 0.25° horizontal resolution. Black circles indicate the locations of orographic hotspots that are investigated in this study (see Table 1 for details).



**Figure 2.** Multi-annual seasonal means (April – October in 2003 – 2014) of ERA-Interim horizontal winds at the AIRS observational level (3 hPa; top) and at low level (750 hPa; bottom). White contour lines appear at levels of 10 m/s for the 3 hPa layer and 5 m/s for the 750 hPa layer. Black circles indicate the locations of orographic hotspots that are investigated in this study (see Table 1 for details).



**Figure 3.** Maps of $4.3\,\mu$m brightness temperature perturbations from individual AIRS/Aqua satellite orbits illustrate stratospheric gravity wave activity at selected orographic hotspots. Red boxes indicate the eastern und western boxes used to detect gravity wave activity.







**Figure 4.** Histograms of 4.3 $\mu$m brightness temperature variances in eastern and western boxes at selected hotspots. Increased numbers of events in the eastern box indicate orographic wave activity. The ratio $n_e/n_w$ refers to the numbers of events below and above the identity line, respectively. Note that the total number of events depends on the number of satellite overpasses, which increases for high latitudes.





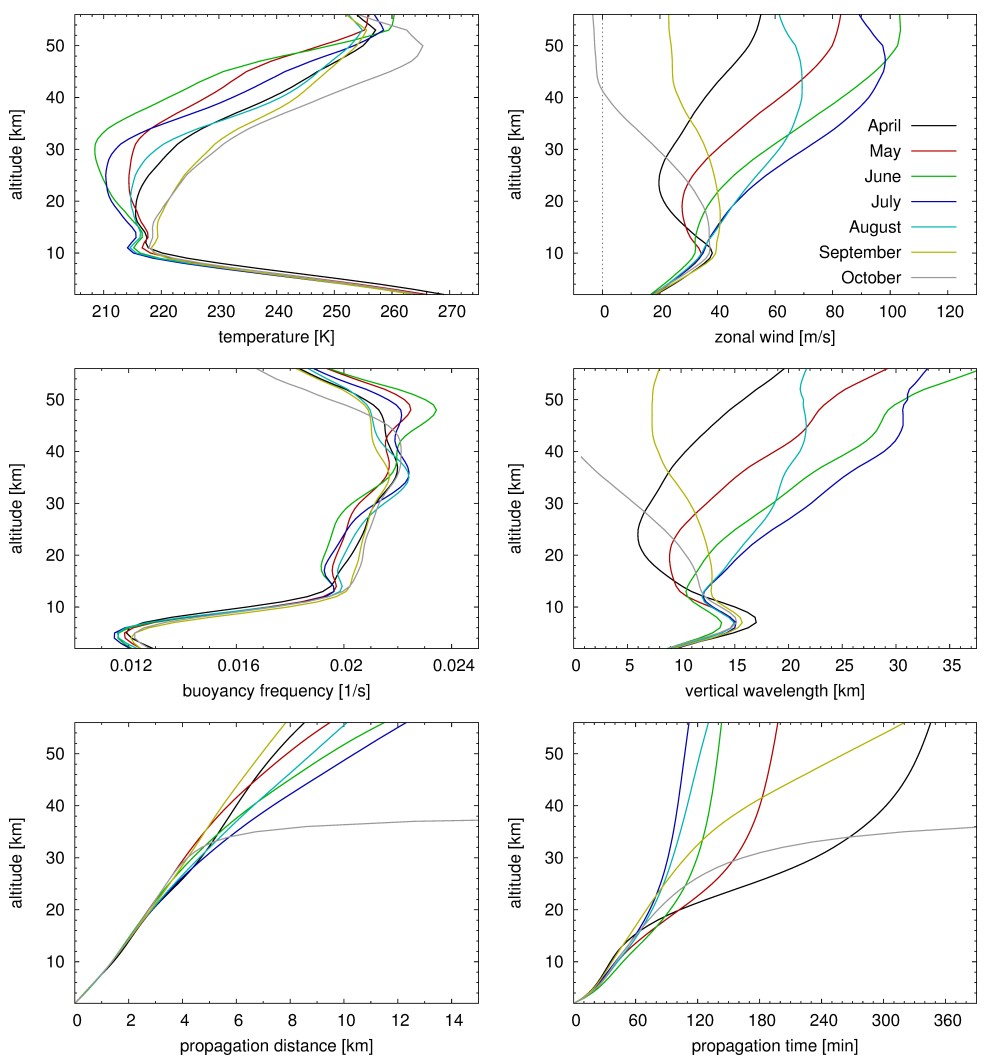

**Figure 5.** Results of 2-D raytracing calculations (middle and bottom) based on ERA-Interim monthly mean temperature and zonal wind profiles (top) at the Kerguelen Islands. A stationary wave with 100 km horizontal wavelength was assumed. The launch level was set to 2 km.





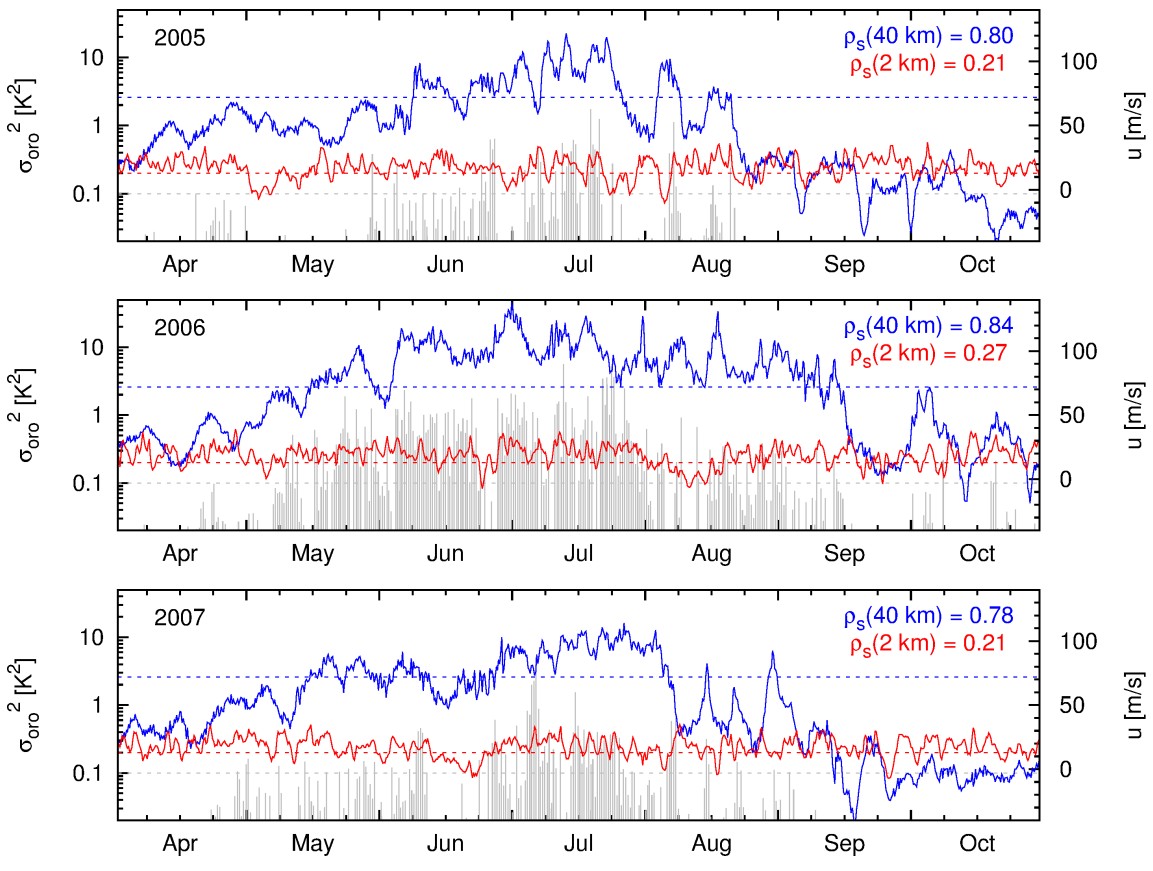

**Figure 6.** Time series of AIRS 4.3 $\mu$m brightness temperature variance differences (gray) and ERA-Interim zonal winds at 2 km (red) and 40 km (blue) log-pressure altitude from 1 April to 31 October in 2005 (top), 2006 (middle), and 2007 (bottom) at the Kerguelen Islands. Dotted lines indicate the 0.1 K$^2$ threshold used to detect orographic gravity waves and zonal wind levels of 13 and 72 m/s used to predict mountain wave events in the AIRS observations.





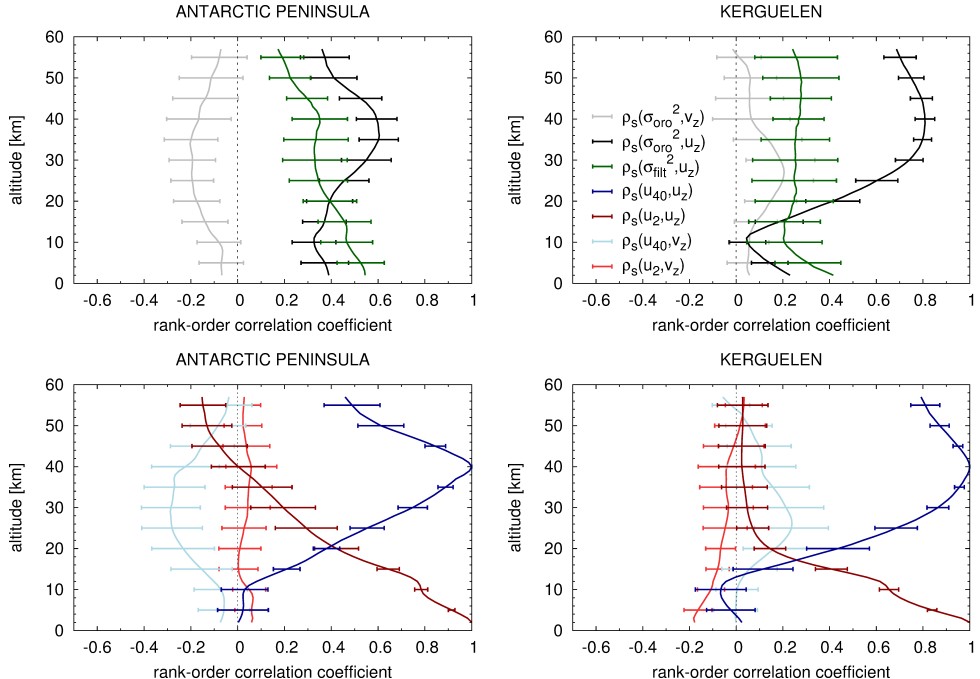

**Figure 7.** Top: Interannual mean and standard deviation of rank-order correlation coefficients of AIRS 4.3 $\mu$m brightness temperature variance differences and ERA-Interim zonal winds (black) and meridional winds (gray) at different altitudes. Green curves show correlation coefficients restricted to cases with 40 km zonal winds exceeding thresholds of $44\,\text{m s}^{-1}$ at the Antarctic Peninsula and $72\,\text{m s}^{-1}$ at Kerguelen Islands, respectively. Bottom: Rank-order correlations of 40 km (blue) and 2 km (red) zonal winds with zonal wind (dark colors) and meridional winds (light colors) at different altitudes.





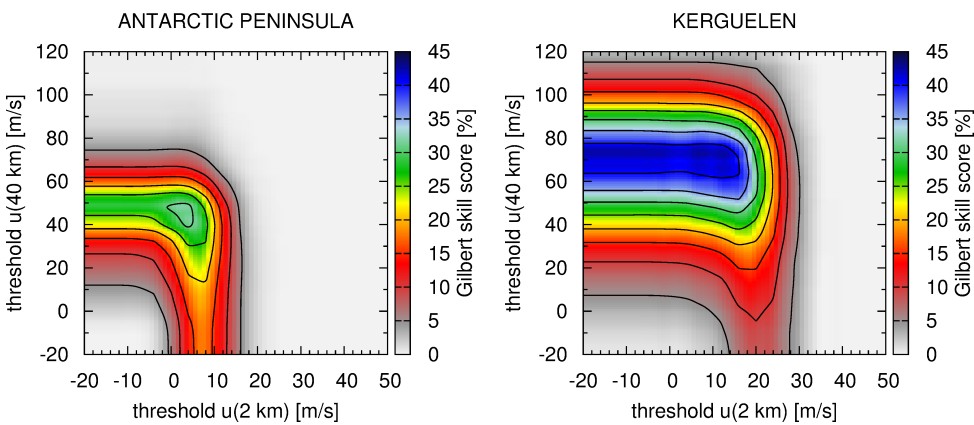

**Figure 8.** Gilbert skill scores of the prediction model for mountain wave events at the Antarctic Peninsula and Kerguelen for different zonal wind thresholds at 2 and 40 km altitude.



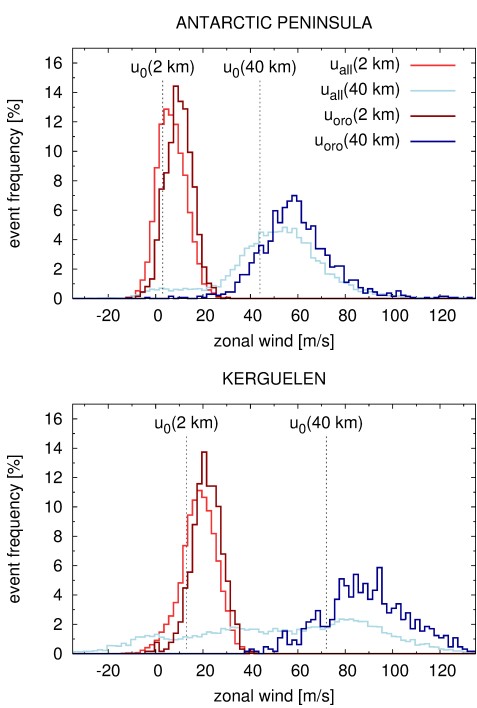

**Figure 9.** Histograms of ERA-Interim zonal winds at 2 km (red) and 40 km (blue) altitude during April – October 2003 – 2014 at the Antarctic Peninsula and Kerguelen. Light colored curves show data for all satellite overpasses. Dark colored curves show data only for overpasses with orographic wave events. Dotted lines indicate the zonal wind thresholds of the mountain wave prediction model.



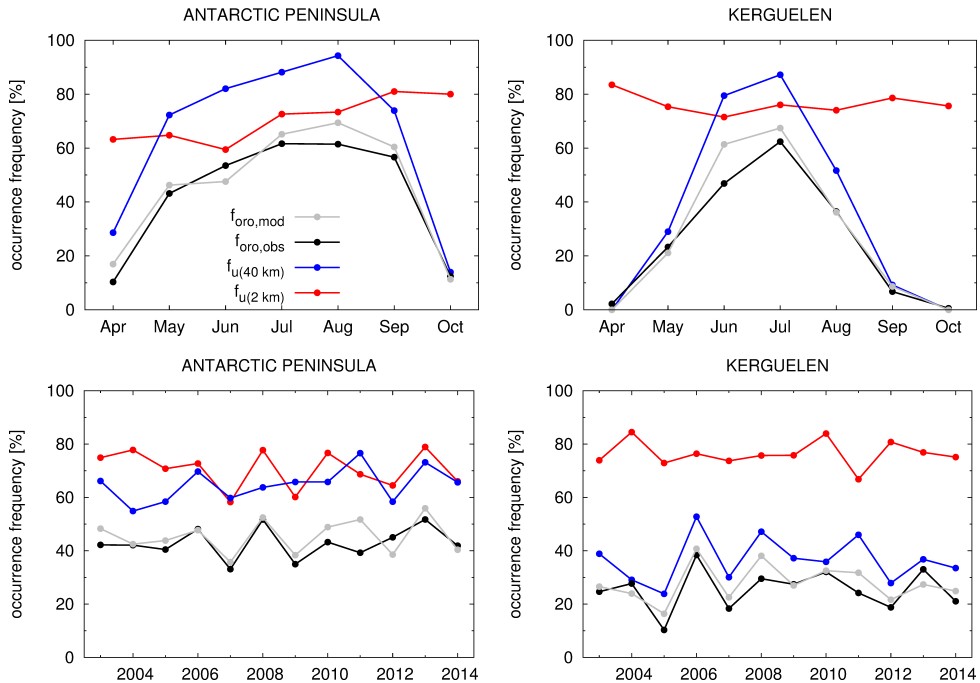

**Figure 10.** Monthly (top) and yearly (bottom) variability of orographic wave activity during April – October in 2003 – 2014 at the Antarctic Peninsula and Kerguelen Islands. Time series show occurrence frequencies of orographic waves from AIRS observations (black) and the mountain wave prediction model (gray). Also shown are occurrence frequencies of the zonal winds at the 2 km (red) and 40 km (blue) levels exceeding the prediction model thresholds.