# Peer review of "Stratospheric gravity waves at southern hemisphere orographic hotspots: 2003 – 2014 AIRS/Aqua observations"

_Atmospheric Chemistry and Physics, 2016_

## Referee Comment (RC1) · A. Dörnbrack (Referee) · 7 Jun 2016

Review of

Stratospheric gravity waves at southern hemisphere orographic hotspots: 2003-2014 AIRS/Aqua observations

by L. Hoffmann, A. W. Grimsdell, and M. J. Alexander

This is a wonderful, I would say perfect paper documenting the stratospheric gravity wave activity (esp. orographic gravity waves) at selected hot spots of the southern hemisphere from 12 years AIRS data. Additionally, the authors also correlate the re-trieved stratospheric mountain wave activity with the horizontal wind components taken

at 2 km and near the AIRS observational level at 40 km, respectively. The methodology is sound, the presentation is very clear, precise, and comprehensive, the figures illustrate and exemplify the necessary information from the observations and meteorological analyses, and the tables add substantial material from a profound statistical analysis of the data. Therefore, the paper can be published in the present form.

Here, only a few thoughts I got while reading the manuscript:

- the enhanced correlation of the observed stratospheric gravity wave activity with the stratospheric winds was also found recently by the study of Kaifler et al. (2015)

Kaifler, B., N. Kaifler, B. Ehard, A. Dörnbrack, M. Rapp, and D. C. Fritts (2015), Influences of source conditions on mountain wave penetration into the stratosphere and mesosphere, Geophys. Res. Lett., 42, 9488–9494, doi:10.1002/2015GL066465.

Additionally, I found it remarkable that the u_0(2 m) threshold for New Zealand (Table 2) lies almost exactly in the range of tropospheric winds for which Kaifler et al. found the largest gravity wave energies at mesospheric altitudes (Fig. 5a) supporting the findings of the present paper!

- I was wondering about the short vertical propagation times and short horizontal propagation distances. Most of the selected examples in Fig. 3 show a much longer horizontal spread of the waves. I always thought that these wave fronts are essentially due to hydrostatic mountain waves in the rotating regime, i.e. due to inertia-gravity waves (see Gill, 1980, p 260) which would imply longer horizontal wave lengths and longer vertical propagation times. But, you might have another explanation.

- the simple mountain wave prediction model used by the authors (very nice!) reminded me on my own first attempt to quantify stratospheric gravity wave above Scandinavia for one winter month based on quite similar criteria (Dörnbrack, A., M. Leutbecher, J. Reichardt, A. Behrendt, K.-P. Müller, and G. Baumgarten, 2001: Relevance of mountain wave cooling over Scandinavia: Mesoscale dynamics and observations for January

1997. J. Geophys. Res., 106, 1569-1581. However, in the present model the wind turning with height is not considered which is clearly understandable for the southern hemispheric conditions without much planetary wave activity diverting the stratospheric winds from nearly pure westerlies!

AD
* * *

---

## Referee Comment (RC2) · Anonymous Referee #2 · 14 Jun 2016

Review comments for "Stratospheric gravity waves at southern hemisphere orographic hotspots: 2003-2014 AIRS/Aqua observations" by Hoffmann et al.

This manuscript develops a novel method ("two-box" method) to detect orographic gravity waves (OGWs) that are associated with Southern Ocean hotspots from AIRS images. The orographic sources are delineated from all other GW cases using this method, and the relationship between the occurrence frequency of OGWs and the background wind at the generation level and the observational level, and the terrain orientation and altitude are studied. A simple deterministic model is proposed to predict the OGW occurrence using zonal wind threshold at the generation and observational levels (750 and 3 hPa, respectively). This model can roughly capture the interannual

variability of the observed GW time series.

This paper is in general well written, the logic flow is clear, natural and fluent. The two box method is a novel way to disentangle OGWs from other types of GWs. The deterministic model is a simple yet effective way to capture potential OGW events that would therefore facilitate future analysis with huge amount of satellite images. This paper worth a final publication as it has good quality of scientific outcomes.

Having said above positive points, I do have a few major issues related to the content/presentation. They are: (1) The "prediction" model is actually not a forecast model, at least on traditional sense. It is more or less a deterministic model that given the lower and upper level wind condition, you can tell how possible an OGW event to occur, but you cannot forecast when and where it occurs. Or to state it in another way, this model does not have a term of (t) and (t+dt) in it, where "t" is time. So please clarify this point clearly in the paper, and modify the wording accordingly.

(2) Another question I have with the "prediction" model is that why do you use this Gill-bert skill score (GSS) to construct the 2-dimensional Probability Density Function, not the averaged variance (i.e., $\sigma^2_{oro}$ ) within the designated boxes? Also, there is no introduction or reference to this GSS, which I have no idea how it is calculated or what physical quantity it can represent.

(3) For the two box method, the authors do play around with the threshold to check its sensitivity, and it turns out it is indeed sensitive to the threshold. Have you also checked the lower threshold (e.g., $0.05K^2$ )? How do you trade-off the detection rate and false alarm rate? In other words, the standard to determine a good threshold is not stated clearly in this paper. I think this part (the last paragraph on Page 6) requires more details and more sensitivity study to determine the best threshold. Also, the box size is really largely dependent on topography size and nearby surroundings. Can you summarize a more generalized way if possible?

(4) Since the number of rays are very limited for the ray-tracing experiment, the critical

level in October is probably just case-by-case. Besides, this is a 2D ray-tracing that has many limitations. Other than illustrating the wind effect on shifting the vertical wavelength toward AIRS-favorable window, I don't see a particular reason of including the 1st paragraph on Page 9 and Fig. 5. It's rather distracted of the main topic of this paper. The linear wave theory presented on Page 8 is pretty straightforward. I suggest deleting the 1st paragraph on Page 9 and Fig. 5.

(5) Instead, since all the factors that play a role in the simple "prediction" model has been included in the GCM OGWD parameterization, I believe. What are the differences? Can your findings shed light on improving the model parameterization? Can your simple model be used to study inter-annual variability? Can you elaborate more on the value of your model?

---

## Referee Comment (RC3) · Anonymous Referee #3 · 23 Jun 2016

**A review report of the manuscript entitled "Stratospheric gravity waves at southern hemisphere orographic hotspots: 2003 – 2014 AIRS/Aqua observations" by L. Hoffmann, A. W. Grimsdell, and M. J. Alexander**

This paper describes a concise and wise method to detect orographic gravity waves from hotspots distributed in the southern hemisphere middle and high latitudes by comparing gravity wave signature in two box areas windward and leeward of the hotspots observed by AIRS. The analyzed period covers about 12 years providing sufficient significance for the statistical results. In addition, based on careful correlation analyses, the authors made a simple model using thresholds for lower and upper level wind which predicts the occurrence of gravity waves detected by AIRS observations with a high score. This model is useful for future extended studies using AIRS observations such as wave flux estimate including geographic and interannual variability. Contents of this paper are quite interesting, well organized and hence has a significant value for publication. However, there are a few points which the authors may need to consider for improvement of the manuscript before publication. Thus, I recommend its publication in Atmospheric Chemistry and Physics after minor revision. Detailed comments are listed below.

**Comments:**

1. p.6 ll. 8-9: Please describe how orographic wave events are detected by visual inspection. Even visual inspection unwittingly uses some conditions to identify the orographic waves such as wavy phase structures and/or strong amplitudes. Are there any common characteristics in the cases when the two-box method failed and visual inspection succeeded in the detection?

2. p.10, ll. 29-30: Please describe the vertical wavelengths corresponding to the zonal wind thresholds obtained using (2). Are they close to the minimum vertical wavelength detectable by AIRS?

3. p.11 ll. 23-32: There may be one more factor influencing POD, that is horizontal wavelengths of gravity waves excited at each hotspot. AIRS detects horizontal wavelengths of about 100 km (p.9 l.5). However, small islands may generate gravity waves with small horizontal wavelengths which are hardly detectable by AIRS. In addition, such small-horizontal scale waves may not be able to propagate through strong westerly wind in the stratosphere. If we include a non-hydrostatic effect, the dispersion relation of gravity waves becomes

$$m^2 = \frac{N^2}{U^2} - k^2.$$

Orographic gravity waves generated by small horizontal-scale islands should have large $k$. The above dispersion relation formula indicates that $m^2$ can be negative in quite large $U$. In such condition, the waves hardly propagate upward and reflect downward at the level where

$\frac{N^2}{U^2} = k^2$. Does not this mechanism happen for the hotspots with relatively low score?

4. References in section 1 seem not sufficient. Suggested references are as follows.

a) p.2, ll.2-3: For orographic generation:

Aircraft observations:

- D. K. Lilly and P. J. Kennedy (1973), Observations of a Stationary Mountain Wave and its Associated Momentum Flux and Energy Dissipation, Journal of the Atmospheric Sciences, 30, 1135-1152

MST/ST radar observations:

- Ecklund, W. L., K. S. Gage, G. D. Nastrom, and B. B. Balsley (1986), A Preliminary Climatology of the Spectrum of Vertical Velocity Observed by Clear-Air Doppler Radar. Journal of Climate and Applied Meteorology, 25, 885-892.
- Sato, K. (1990), Vertical Wind Disturbances in the Troposphere and Lower Stratosphere Observed by the MU Radar, Journal of the Atmospheric Sciences, 47, 2803-2817.
- Worthington, R. M., and L. Thomas (1996), Radar measurements of critical-layer absorption in mountain waves. Quart. J. Roy. Meteor. Soc., 122, 1263−1282.
- Minamihara, Y., K. Sato, M. Kohma, and M. Tsutsumi (2016), Characteristics of Vertical Wind Fluctuations in the Lower Troposphere at Syowa Station in the Antarctic Revealed by the PANSY Radar, SOLA, 12, 116−120, doi:10.2151/sola.2016-026

p.2, ll.4-5: For adjustment of flow imbalance:

- Plougonven, R., and F. Zhang (2014), Internal gravity waves from atmospheric jets and fronts, Rev. Geophys., 52, 33–76, doi:10.1002/2012RG000419.

p.2, ll.9-10: For general discussion of gravity wave source in the summer and winter hemispheres:

- Sato, K., S. Watanabe, Y. Kawatani, Y. Tomikawa, K. Miyazaki, and M. Takahashi (2009), On the origins of mesospheric gravity waves. Geophys. Res. Lett., 36, L19801, doi:10.1029/2009GL039908.

p.2, l.11: Orographic hotpots in the southern ocean:

- Wu, D. L., P. Preusse, S. D. Eckermann, J. H. Jiang, M. T. Juarez, L. Coy, D. Y. Wang (2006), Remote sounding of atmospheric gravity waves with satellite limb and nadir techniques, Adv. Space Res., 37, 2269–2277, 2006

p.2, l.19: For gravity wave effects on PSCs:

- Alexander, S. P., A. R. Klekociuk, M. C. Pitts, A. J. McDonald, and A. Arevalo-Torres (2011), The effect of orographic gravity waves on Antarctic polar stratospheric cloud occurrence and composition, J. Geophys. Res., 116, D06109, doi:10.1029/2010JD015184.

- Kohma, M., and K. Sato (2011), The effects of atmospheric waves on the amounts of polar stratospheric clouds. Atmos. Chem. Phys., 11, 11535-11552. doi:10.5194/acp-11-11535-2011.

---

## Author Comment (AC1) · 5 Jul 2016

**Reply to review comments**

We thank the reviewers for the time and efforts spent on the manuscript. Please find our point-by-point replies to the review comments below (colored in blue). A revised manuscript with tracked changes was uploaded.

**Reviewer #1 (Andreas Dörnbrack)**

This is a wonderful, I would say perfect paper documenting the stratospheric gravity wave activity (esp. orographic gravity waves) at selected hot spots of the southern hemisphere from 12 years AIRS data. Additionally, the authors also correlate the retrieved stratospheric mountain wave activity with the horizontal wind components taken at 2 km and near the AIRS observational level at 40 km, respectively. The methodology is sound, the presentation is very clear, precise, and comprehensive, the figures illustrate and exemplify the necessary information from the observations and meteorological analyses, and the tables add substantial material from a profound statistical analysis of the data. Therefore, the paper can be published in the present form.

Here, only a few thoughts I got while reading the manuscript:

- the enhanced correlation of the observed stratospheric gravity wave activity with the stratospheric winds was also found recently by the study of Kaifler et al. (2015)

Kaifler, B., N. Kaifler, B. Ehard, A. Dörnbrack, M. Rapp, and D. C. Fritts (2015), Influences of source conditions on mountain wave penetration into the stratosphere and mesosphere, Geophys. Res. Lett., 42, 94889494, doi:10.1002/2015GL066465.

Additionally, I found it remarkable that the $u_0(2\,\mathrm{km})$ threshold for New Zealand (Table 2) lies almost exactly in the range of tropospheric winds for which Kaifler et al. found the largest gravity wave energies at mesospheric altitudes (Fig. 5a) supporting the findings of the present paper!

We added a brief discussion of the findings of the study of *Kaifler et al.* (2015) to the conclusions sections of our paper.

- I was wondering about the short vertical propagation times and short horizontal propagation distances. Most of the selected examples in Fig. 3 show a much longer horizontal spread of the waves. I always thought that these wave fronts are essentially due to hydrostatic mountain waves in the rotating regime, i.e. due to inertia-gravity waves (see Gill, 1980, p 260) which would imply longer horizontal wave lengths and longer vertical propagation times. But, you might have another explanation.

Following a comment of Reviewer #2, we removed the results of the ray-tracing calculations from the paper. These results are not needed for the discussion.

- the simple mountain wave prediction model used by the authors (very nice!) reminded me on my own first attempt to quantify stratospheric gravity wave above Scandinavia for one winter month based on quite similar criteria (Dörnbrack, A., M. Leutbecher, J. Reichardt, A. Behrendt, K.-P. Müller, and G. Baumgarten, 2001: Relevance of mountain wave cooling over Scandinavia: Mesoscale dynamics and observations for January 1997. J. Geophys. Res., 106, 1569-1581. However, in the present model the wind turning with height is not considered which is clearly understandable for the southern hemispheric conditions without much planetary wave activity diverting the stratospheric winds from nearly pure westerlies!

Our model is indeed similar to the one proposed by *Dörnbrack et al.* (2001) and we added a brief discussion and the reference to the introduction of our paper.

**Reviewer #2**

This manuscript develops a novel method ("two-box" method) to detect orographic gravity waves (OGWs) that are associated with Southern Ocean hotspots from AIRS images. The orographic sources are delineated from all other GW cases using this method, and the relationship between the occurrence frequency of OGWs and the background wind at the generation level and the observational level, and the terrain orientation and altitude are studied. A simple deterministic model is proposed to predict the OGW occurrence using zonal wind threshold at the generation and observational levels (750 and 3 hPa, respectively). This model can roughly capture the interannual variability of the observed GW time series.

This paper is in general well written, the logic flow is clear, natural and fluent. The two box method is a novel way to disentangle OGWs from other types of GWs. The deterministic model is a simple yet effective way to capture potential OGW events that would therefore facilitate future analysis with huge amount of satellite images. This paper worth a final publication as it has good quality of scientific outcomes.

Having said above positive points, I do have a few major issues related to the content/presentation. They are: (1) The "prediction" model is actually not a forecast model, at least on traditional sense. It is more or less a deterministic model that given the lower and upper level wind condition, you can tell how possible an OGW event to occur, but you cannot forecast when and where it occurs. Or to state it in another way, this model does not have a term of (t) and (t+dt) in it, where "t" is time. So please clarify this point clearly in the paper, and modify the wording accordingly.

We use the term "prediction" according to its definition in a statistical sense (e. g., *Cox*, 2006). Prediction provides a means of transferring knowledge about a sample of a population, and to other related populations, which is not necessarily the same as prediction over time. When information is transferred across time, often to

specific points in time, the process is known as forecasting. We revised the manuscript to remove references to the term "forecast" to avoid misinterpretation.

(2) Another question I have with the "prediction" model is that why do you use this Gilbert skill score (GSS) to construct the 2-dimensional Probability Density Function, not the averaged variance (i.e., $\sigma_{oro}^2$) within the designated boxes? Also, there is no introduction or reference to this GSS, which I have no idea how it is calculated or what physical quantity it can represent.

We added a more detailed description of the GSS to clarify: "In particular, we analyze the Gilbert skill score (GSS), which is also known as 'equitable threat score' (*Schaefer*, 1990; *Wilks*, 2011). This is a standard verification method for dichotomous (yes/no) model predictions. It takes into account the probability of detection (POD) and the false alarm rate (FAR) of the model and is adjusted for hits associated with random chance." Trying to predict gravity wave variances ($\sigma_{oro}^2$) with a model is an interesting topic. However, our model focuses on predicting occurrence frequencies of gravity wave events. The GSS is focusing on verification of yes/no predictions and is therefore well suited to validate our type of model.

(3) For the two box method, the authors do play around with the threshold to check its sensitivity, and it turns out it is indeed sensitive to the threshold. Have you also checked the lower threshold (e.g., $0.05\,\mathrm{K}^2$)? How do you trade-off the detection rate and false alarm rate? In other words, the standard to determine a good threshold is not stated clearly in this paper. I think this part (the last paragraph on Page 6) requires more details and more sensitivity study to determine the best threshold. Also, the box size is really largely dependent on topography size and nearby surroundings. Can you summarize a more generalized way if possible?

The last paragraph of Sect. 3 provides an extensive discussion on how we selected the variance threshold $\sigma_{oro}^2$. We rephrased it a bit to try to make our approach more clear. A threshold of $0.05\,\mathrm{K}^2$ is outside the range we consider reasonable for this analysis. We had tested an intermediate value of $0.3\,\mathrm{K}^2$, which provides GSS values in between the results for 0.1 and $1\,\mathrm{K}^2$ (not reported in the paper). This led us to the conclusion that $0.1\,\mathrm{K}^2$ is the optimum choice within the considered range. The trade-off between POD and FAR is fixed in the definition of the GSS. Please note that the verification uses data of *Alexander and Grimsdell* (2013) as a reference. The reference data are determined by visual inspection, use a different set of AIRS channels, and are available only for a subset of hotspots and years. Therefore the reference data can provide guidance, but cannot be used to perform fine-tuning to determine the "best" choice of the variance threshold. The motivation for choosing the box sizes individually is provided in Sect. 3, 2nd paragraph.

(4) Since the number of rays are very limited for the ray-tracing experiment, the critical level in October is probably just case-by-case. Besides, this is a 2D ray-tracing that has many limitations. Other than illustrating the wind effect on shifting the vertical wavelength toward AIRS-favorable window, I dont see a particular reason of including the 1st

paragraph on Page 9 and Fig. 5. Its rather distracted of the main topic of this paper. The linear wave theory presented on Page 8 is pretty straightforward. I suggest deleting the 1st paragraph on Page 9 and Fig. 5.

We agree and removed the results of the ray-tracing calculations from the paper.

(5) Instead, since all the factors that play a role in the simple "prediction" model has been included in the GCM OGWD parameterization, I believe. What are the differences? Can your findings shed light on improving the model parameterization? Can your simple model be used to study inter-annual variability? Can you elaborate more on the value of your model?

In the introduction we pointed out "The main purpose of our model is to provide a means of separating upper level wind effects, like the observational filter, from low level effects, like those related to the gravity wave sources. This will allow the model to be used to discuss whether waves are likely present or affecting the atmosphere even though they are only weakly observed or invisible in the AIRS observations." The model is not intended to be a replacement for more sophisticated orographic gravity wave dag parametrization schemes used in general circulation models.

**Reviewer #3**

This paper describes a concise and wise method to detect orographic gravity waves from hotspots distributed in the southern hemisphere middle and high latitudes by comparing gravity wave signature in two box areas windward and leeward of the hotspots observed by AIRS. The analyzed period covers about 12 years providing sufficient significance for the statistical results. In addition, based on careful correlation analyses, the authors made a simple model using thresholds for lower and upper level wind which predicts the occurrence of gravity waves detected by AIRS observations with a high score. This model is useful for future extended studies using AIRS observations such as wave flux estimate including geographic and interannual variability. Contents of this paper are quite interesting, well organized and hence has a significant value for publication. However, there are a few points which the authors may need to consider for improvement of the manuscript before publication. Thus, I recommend its publication in Atmospheric Chemistry and Physics after minor revision. Detailed comments are listed below.

Comments:

1. p.6, ll. 8-9: Please describe how orographic wave events are detected by visual inspection. Even visual inspection unwittingly uses some conditions to identify the orographic waves such as wavy phase structures and/or strong amplitudes. Are there any common characteristics in the cases when the two-box method failed and visual inspection succeeded in the detection?

To clarify we added: "In order to identify orographic waves in as objective and consistent a manner as possible, the visual inspections follows criteria defined by *Alexander and Grimsdell* (2013). In particular, there must be a clear difference in the wave pattern near the hotspot to distinguish orographic waves from waves from other sources, i. e., the location of the hotspot should be clearly indicated by the position of the wave pattern. Furthermore, if the observation includes both an orographic wave and a larger-scale background wave pattern, there must be a distinct change in the pattern directly adjacent to the hotspot." We were not able to identify common characteristics of the cases when the two-box method failed and visual inspection was successful. The reasons are manifold.

2. p.10, ll. 29-30: Please describe the vertical wavelengths corresponding to the zonal wind thresholds obtained using (2). Are they close to the minimum vertical wavelength detectable by AIRS?

We added: "For zonal waves these thresholds correspond to vertical wavelengths of 14 and 23 km, which are close to or well above the AIRS detection limit, respectively." Please note that the orientation of the wind may also play a role, in particular for the Antarctic Peninsula.

3. p.11, ll. 23-32: There may be one more factor influencing POD, that is horizontal wavelengths of gravity waves excited at each hotspot. AIRS detects horizontal wavelengths of about 100 km (p.9 l.5). However, small islands may generate gravity waves with small horizontal wavelengths which are hardly detectable by AIRS. In addition, such small-horizontal scale waves may not be able to propagate through strong westerly wind in the stratosphere. If we include a non-hydrostatic effect, the dispersion relation of gravity waves becomes

$$m^2 = \frac{N^2}{U^2} - k^2.$$

Orographic gravity waves generated by small horizontal-scale islands should have large $k$. The above dispersion relation formula indicates that $m^2$ can be negative in quite large $U$. In such condition, the waves hardly propagate upward and reflect downward at the level where $\frac{N^2}{U^2} = k^2$. Does not this mechanism happen for the hotspots with relatively low score?

We agree that the terrain horizontal extent may also play an important role for the characteristics of the wave emitted from the orographic sources. We added this factor to the list provided in the manuscript.

4. References in section 1 seem not sufficient. Suggested references are as follows.

a) p.2, ll.2-3: For orographic generation:

Aircraft observations:

- D. K. Lilly and P. J. Kennedy (1973), Observations of a Stationary Mountain Wave

and its Associated Momentum Flux and Energy Dissipation, Journal of the Atmospheric Sciences, 30, 1135-1152

MST/ST radar observations:

- Ecklund, W. L., K. S. Gage, G. D. Nastrom, and B. B. Balsley (1986), A Preliminary Climatology of the Spectrum of Vertical Velocity Observed by Clear-Air Doppler Radar. Journal of Climate and Applied Meteorology, 25, 885-892.

- Sato, K. (1990), Vertical Wind Disturbances in the Troposphere and Lower Stratosphere Observed by the MU Radar, Journal of the Atmospheric Sciences, 47, 2803-2817.

- Worthington, R. M., and L. Thomas (1996), Radar measurements of critical-layer absorption in mountain waves. Quart. J. Roy. Meteor. Soc., 122, 12631282.

- Minamihara, Y., K. Sato, M. Kohma, and M. Tsutsumi (2016), Characteristics of Vertical Wind Fluctuations in the Lower Troposphere at Syowa Station in the Antarctic Revealed by the PANSY Radar, SOLA, 12, 116-120, doi:10.2151/sola.2016-026

p.2, ll.4-5: For adjustment of flow imbalance:

- Plougonven, R., and F. Zhang (2014), Internal gravity waves from atmospheric jets and fronts, Rev. Geophys., 52, 3376, doi:10.1002/2012RG000419.

p.2, ll.9-10: For general discussion of gravity wave source in the summer and winter hemispheres:

- Sato, K., S. Watanabe, Y. Kawatani, Y. Tomikawa, K. Miyazaki, and M. Takahashi (2009), On the origins of mesospheric gravity waves. Geophys. Res. Lett., 36, L19801, doi:10.1029/2009GL039908.

p.2, l.11: Orographic hotpots in the southern ocean:

- Wu, D. L., P. Preusse, S. D. Eckermann, J. H. Jiang, M. T. Juarez, L. Coy, D. Y. Wang (2006), Remote sounding of atmospheric gravity waves with satellite limb and nadir techniques, Adv. Space Res., 37, 22692277, 2006

p.2, l.19: For gravity wave effects on PSCs:

- Alexander, S. P., A. R. Klekociuk, M. C. Pitts, A. J. McDonald, and A. Arevalo-Torres(2011), The effect of orographic gravity waves on Antarctic polar stratospheric cloud occurrence and composition, J. Geophys. Res., 116, D06109, doi:10.1029/2010JD015184.

- Kohma, M., and K. Sato (2011), The effects of atmospheric waves on the amounts of polar stratospheric clouds. Atmos. Chem. Phys., 11, 11535-11552. doi:10.5194/acp-11-11535-2011.

We added these references.

[revised manuscript text omitted]